# 'More tight-less tight' Patterns in the Climatic Niche Evolution of *Gymnocalycium* (Cactaceae): Were Pleistocene Glaciations a Prelude?

Rahul Raveendran Nair[1,2]*, Alicia N. Sérsic[2], Pablo H. Demaio[3], Solana B. Perotti[4], Diego E. Gurvich[2,5]

1 Biodiversity Institute, University of Kansas, Lawrence, Kansas, United States of America, 2 Instituto Multidisciplinario de Biología Vegetal (Universidad Nacional de Córdoba-CONICET), Córdoba, Argentina, 3 Cátedra de Botánica Sistemática, Facultad de Ciencias Agrarias, Universidad Nacional de Catamarca, Catamarca, Argentina, 4 Cátedra de Ecología, Facultad de Ciencias Exactas y Naturales, Universidad Nacional de Catamarca, Catamarca, Argentina, 5 Cátedra de Morfología Vegetal, Facultad de Ciencias Exactas, Físicas y Naturales, Universidad Nacional de Córdoba, Córdoba, Argentina

* rahulraveendran@ku.edu

## Abstract

Cacti have undergone some of the fastest diversification events in the plant kingdom despite their slow growth rates and extended generation times. This rapid evolution may be driven by intense ecological interactions. Here we tested, for the first time, the evolutionary dynamics of the ecological niches of *Gymnocalycium* species focusing on two key environmental factors: temperature and precipitation. To explore patterns of niche conservatism and/or evolution and identify major contributing factors, we reconstructed ancestral niches associated with these climatic dimensions using the binned ancestral range coding method. Our findings reveal that (1) the climatic-niches of narrow-ranged *Gymnocalycium* species are not highly conserved across the phylogeny (i.e., niches are evolving), (2) the evolutionary dynamics of thermal and precipitation niches across the *Gymnocalycium* phylogeny do not follow similar patterns, (3) a bioregion-specific pattern of niche evolution exists, and (4) the Early–Middle Pleistocene glaciations (i.e., GPG and three Post-GPG phases) potentially drove the patterns of lineage divergence in *Gymnocalycium* species, triggering the evolution of climatic niches. These results suggest that (i) *Gymnocalycium* species with fascicular roots may require special attention for conservation, (ii) in a warming climate, the species distributed in the South American transition zone, South Brazilian dominion, and Chacoan dominion may face serious risks, and (iii) the relatively 'more tight-less tight' pattern in conserving the precipitation and temperature niches could be a strategy for conserving the critical variable at the expense of the other. This study has not only provided valid insights into the evolutionary history of *Gymnocalycium* species but also highlights the importance of conservation efforts, essential to protect these species.

**Data availability statement:** All relevant data are within the manuscript and its Supporting Information files.

**Funding:** The financial support received from the British Cactus and Succulent Society (BCSS, UK) for the project titled "Evolutionary history of four Gymnocalycium species of the Sierras Pampeanas at intra- and interspecific level" helped in part to complete this study. Details of the project can be found at: https://bcss.org.uk/evolutionary-history-of-four-gymnocalycium-species-of-the-sierras-pampeanas-at-intra-and-interspecific-level/. In addition, this research received financial support from FONCyT (PICT 2016-0077) and SECyT-FCEFyN-UNC (Universidad Nacional de Córdoba, Argentina). The funders had no role in study design, data collection and analysis, decision to publish, or preparation of the manuscript.

**Competing interests:** The authors have declared that no competing interests exist.

## Introduction

In evolutionary studies, a strong link between environmental conditions and speciation in Cactaceae has been established, with environmental factors exerting selective pressures on floral traits [1], maintaining species boundaries [2], and influencing lineage diversification [3,4]. The concept of species' niches is central in understanding the patterns and boundaries of species' response to the environment [5,6]. Species' fundamental niche—the full suite of abiotic conditions supporting its existence without biotic interactions—evolves over time [7–10]. Recent studies have highlighted the evolutionary dynamics of fundamental niches in diverging lineages, enriching our understanding of biogeographic patterns and evolutionary histories [8,9,11–16]. Despite the possible influence of environmental selective pressures on the evolution of different cactus life-forms [17], their natural geographic distribution is also largely, but differentially, shaped by various environmental gradients [18]. For instance, low-temperature sensitivity influences the distribution of columnar cacti [17,19,20], high-temperature [21,22] and the variability of summer rainfall [17] control the distribution of globose cacti, summer rainfall trends affect opuntioid cacti [17], high-moisture conditions support epiphytic cacti distribution [23], and seasonal rainfall patterns influence pereskioid cacti [24]. Given the role of environmental gradients in cactus diversification, linking phylogeny and environment offers new perspectives on niche evolution, indicating whether niches are conserved or divergent across phylogenies [9,15]. Contrary to the common perception that cactus species may withstand future climatic fluctuations, as they are heat and drought tolerant, a recent study [25] indicates that climate change could be the primary risk factor, causing the extinction of many cactus species. Hence, understanding the tendency of cactus species to retain their environmental niches across the phylogeny may help the researchers to better predict their ability to cope with changing climates (i.e., adaptation to novel environmental conditions) [13,26,27].

To the best of our knowledge, a detailed study on the evolutionary dynamics of climatic niches, taking ancestral niches into account and including a dated phylogeny, has never been attempted in any genus of the family Cactaceae. In this study, we focus on the genus *Gymnocalycium* Pfeiff. ex Mittler (Cactaceae), a group of globose cacti species, native to southern South America, that includes around 50 species [28–30]. The currently known distribution of *Gymnocalycium* species in South America spans a broad range, extending from 16.70°S to 45.19°S, and from 53.10°W to 69.30°W, covering five countries: Argentina, Bolivia, Paraguay, Uruguay, and Brazil. It is one of the most important plant genera of the Cactaceae family, as it is species-rich, and is developing a growing interest among both professional and amateur collectors, of succulent or ornamental plants [28,31]. This genus is of particular interest for studying environmental niche evolution across phylogeny due to its (1) high level of endemism [29,31,32], (2) adaptation to heterogenous environmental conditions [29,33–36], and (3) presence at different ecological regions across a range of altitudinal gradients [21,29–32,37,38].

In this study, we tested the evolutionary dynamics of climatic niches across the *Gymnocalycium* phylogeny, asking the following questions.

1. Are the climatic-niches of narrow-ranged *Gymnocalycium* species highly conserved across the phylogeny, indicating niche conservatism?

Various well-established hypotheses, such as niche breadth-range size [39,40], range size-niche breadth [41–43], and range shift-niche breadth [44], that connect niche breadth with the size of geographic ranges, suggest a positive association between niche breadth and geographic range size [45–49]. In a phylogenetic framework, studies on climatic-niche evolution linked with range size suggest that the rate of niche evolution in narrow-ranged species may be lower (i.e., more conserved), compared to wide-ranged species [11,50–52]. However, inter-specific variations in the utilization of environmental space from the total environmental space of the clade may result in the divergence of niches [50]. There are inter-specific differences in the utilization of environmental conditions by the *Gymnocalycium* species. Hence, we expect that the climatic-niches of narrow-ranged *Gymnocalycium* species may not be highly conserved across the phylogeny.

2. Do the evolutionary dynamics of thermal and precipitation niches across the *Gymnocalycium* phylogeny follow similar patterns?

As there are strong selective pressures of temperature and precipitation variables on *Gymnocalycium* species distribution [29,30,33,37,38,53], we expect that the evolutionary dynamics of thermal and precipitation niches across the *Gymnocalycium* phylogeny do not follow similar patterns.

3. Does a bioregion-specific pattern of niche evolution exist with respect to temperature and precipitation dimensions?

Temperature and precipitation variables are central in biogeographic regionalization [54], and the Neotropics have diverse biogeographic regions (bioregions) [55,56]. Hence, we expect a bioregion-specific pattern of niche evolution in *Gymnocalycium*.

4. Did the Early–Middle Pleistocene glaciations (i.e., GPG and three Post-GPG phases) potentially drive the patterns of lineage divergence in *Gymnocalycium* species, resulting in the evolution of temperature and precipitation niches?

Several studies have linked the evolution of climatic niches with lineage divergence events [14,16,57–61]. In South America, the Great Patagonian Glaciation (GPG; 1.0–1.1 Mya) followed by three Middle Pleistocene glaciations (Post-GPG 1, 2 & 3) [62,63] greatly influenced species diversifications and distributions [64–67], as the glacial cycles of advancement and retreat had significant effects on the climate in southern South America (the regions south of 15ºS)[68]. Hence, we expect that the Early–Middle Pleistocene glaciations (i.e., GPG and three Post-GPG phases) played a significant role in lineage divergence in *Gymnocalycium* species, leading to the evolution of temperature and precipitation niches.

## Materials and methods

### Study species and data collection

For the present study, we used 40 *Gymnocalycium* species (S1 Table) that were included in the best inter-specific phylogenetic hypothesis of the *Gymnocalycium* group [69] available to us. The occurrence data of these species were extracted from a variety of sources: field surveys conducted by the authors (RRN, ANS, PHD, SBP, and DEG), online databases such as Global Biodiversity Information Facility (www.gbif.org) [70], iNaturalist (www.inaturalist.org), cactus and succulent field number (www.cl-cactus.com), ecoregistros (www.ecoregistros.org), cactus habitat (www.cactushabitat.com), cactus in habitat (www.cactusinhabitat.org), personal communication (Roberto Kiesling), herbarium collection (CTES herbarium; Instituto de Botánica del Nordeste, Argentina), Documenta Florae Australis (www.darwin.edu.ar/iris), and published literature [71–73]. Documented occurrences with no precise geographic coordinates but with exact locality information were

extrapolated using the Google Earth software to find corresponding geo-coordinates. Each occurrence locality was visually inspected for accuracy using a geographic information system platform (QGIS version 3.36) [74], and questionable records were removed from the dataset. A spatial thinning procedure with a distance threshold of 5.2 km was performed on the dataset to keep only one presence point per pixel in the environmental raster layers (see below for more details). The thinning procedure was executed using the *spThin* R package [75]. The total number of records in the final dataset was 448 (Fig 1); species-wise, a maximum of 54 records for *G. pflanzii* and a minimum of 2 records for *G. chacoense* were present in the dataset.

The bioclimatic raster layers (2.5 arcmin resolution; 19 variables) that serve as the reference of current climate (1979–2013) were downloaded from the PaleoClim database [76], that hosts the upscaled version of high-resolution CHELSA environmental layers [77]. From the climatic dataset, we removed four interactive variables that combine both precipitation and temperature measurements (bio8: mean temperature of wettest quarter, bio9: mean temperature of driest quarter, bio18: precipitation of warmest quarter, and bio19: precipitation of coldest quarter), as these variables are known to possess spatial anomalies [78–82].

## Accessible area hypothesis

Defining the parts of the world that are accessible to species *via* dispersal over relevant periods of time (area **M**) [7,83], with which species' response to environment can be estimated [84], is a crucial component in the characterization of ancestral environmental niches [9]. If niche estimates of a species are derived from the presence data alone (i.e., only from the occupied ranges), it may inflate the estimates of niche change over time which lead to erroneous reconstruction of ancestral niches, as characterization of niches solely based on realized niche is incomplete [8,9]. Incorporating the accessible areas to quantify the utilization of environmental conditions helps to identify cases in which niches estimates are likely to be uncertain [9]. Under the current climate scenarios, the estimation of area **M** was performed using a simulation-based approach [84]; a process that critically considers dispersal parameters (i.e., dispersal kernel and number of dispersal events) and current distributional knowledge [84,85] for each of the *Gymnocalycium* species. The **M** simulation procedure was performed using the grinnell R package (version 0.0.22; https://github.com/fmachados/grinnell.git) [84]. We used a log-normal dispersal kernel for the **M** simulations, as it is best suited for seed dispersal studies [86–89]. Total number of dispersal events was set to 35 (~one dispersal event/year for 35 years, covering the time range of climate data), with two as a maximum number of dispersers that move from each pixel per dispersal event. The parameter (maximum number of dispersers) splits the suitability range (0–1) of a colonized cell into a number of intervals equal to the value of the parameter [84]. In this case, the suitability range (0–1) is divided into two equal intervals: 0.0–0.5 and 0.5–1.0. In each dispersal event, two dispersers will move from a colonized cell if the suitability of that cell exceeds 0.5, and only one disperser will move if the suitability is less than or equal to 0.5. Three different values (0.25, 0.50, and 0.75) were screened for kernel standard deviation (SD) that indicates the spread of dispersal kernel. Parameterization of the **M** simulations is detailed in the supporting information (S2 Table). Finally, the **M**s that were simulated with an SD of 0.50 were chosen for further analyses (Fig 2). This selection was performed by superimposing all **M** polygons on the South American map, and choosing the appropriate **M**s based on visual inspection and expert knowledge. It was not possible to simulate **M**s for eight species due to a low number of records (*G. chacoense, G. glaucum, G. horstii, G. kieslingii, G. oenanthemum, G. robustum, G. stenopleurum,* and *G. uebelmannianum,* and for those species, environmental information was sourced from the occurrence points only.

## Phylogenetic hypothesis and dating

For molecular dating, we performed a Bayesian phylogenetic reconstruction using the aligned DNA sequence matrix (6195-bp) of four plastid markers *viz.*, *atpI-atpH, petL-psbE, trnT-trnL-trnF,* and *trnK-matK,* that was employed earlier for developing the latest available phylogeny of the *Gymnocalycium* group that display inter-specific relationships [69].

**Fig 1. Current distribution.** The distribution of forty species of *Gymnocalycium* included in the present study across five South American countries: Argentina, Brazil, Bolivia, Paraguay, and Uruguay. The shapefiles of countries were sourced from DIVA-GIS (https://diva-gis.org), a free and open-source geographic information system.

The ingroup taxa comprised 40 species of *Gymnocalycium. Matucana polzii*, *Oreocereus celsianus*, *Uebelmannia pectinifera*, and *Stetsonia coryne* were used as outgroup taxa, following Demaio et al. [69]. The appropriate nucleotide substitution model was identified as GTR + G based on the lowest Akaike information criterion (AIC), using the jModel-Test software [90]. To quantify the molecular substitution rates for estimating divergence times, we used an uncorrelated

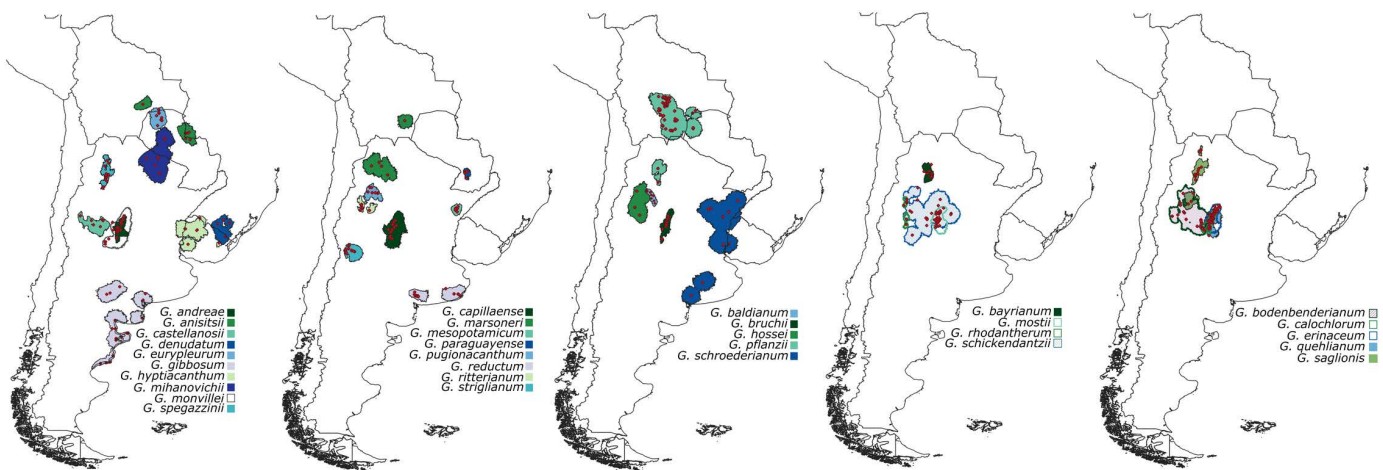

**Fig 2. Accessible areas.** Simulated accessible areas of *Gymnocalycium* species used in this study. The **M** simulations were not successful for eight species: *G. chacoense*, *G. glaucum*, *G. horstii*, *G. kieslingii*, *G. oenanthemum*, *G. robustum*, *G. stenopleurum*, and *G. uebelmannianum*. The red dots represent the known occurrences. The shapefiles of countries were sourced from DIVA-GIS (https://diva-gis.org), a free and open-source geographic information system.

relaxed clock [91] with a log-normal relaxed distribution in combination with Yule tree priors [92]. The time to most recent common ancestor (tMRCA) for the Trichocereeae clade comprising *Gymnocalycium* species, *M. polzii*, and *O. celsianus* was set to 6.12 Mya, following Hernández-Hernández et al. [93]. A root height of 6.57 Mya was set as the divergence time estimate between the Trichocereeae group and the clade comprising *U. pectinifera* and *S. coryne.* We set the length of MCMC chain to $5 \times 10^7$ generations, with parameter values sampled every 5000 generations. The input file was prepared in BEAUti, and the divergence time estimation was performed using BEAST v.1.7.5 [94]. The MCMC output was checked for the convergence of estimated parameters in Tracer v.1.7 [95], and the first 10% of the trees were removed as burn-in [96]. A maximum clade credibility (MCC) tree was constructed by combining the sample trees using TreeAnnotator v.1.10.4 [97], discarding 5000 sampled states. The final tree was visualized in FigTree v.1.4.4 [98].

## Characterization of ancestral niches

As our primary interest was to study the ancestral niche characteristics related to temperature and precipitation dimensions, we used two environmental layers that correspond to mean annual temperature (bio1; hereafter temperature) and annual precipitation (bio12; hereafter precipitation) for further analyses. We selected bio1 and bio12, as these variables reflect the most important aspects of the overall climatic distribution of a species [58,99]. A Binned Ancestral Range (BAR) reconstruction procedure [8,9] that explicitly takes the incomplete characterization of niches (i.e., niche truncation) [8] into consideration was employed independently for these two variables [9,14,16]. Recent years have witnessed a growing interest in using this approach to reconstruct the ancestral niches [9,14,16,100]. Summary statistics of these two variables (i.e., mean, standard deviation, median, minimum and maximum values, and quartiles) based on the accessible areas (**M**s) and the occurrence localities, were estimated separately. This estimation of the full niche ranges was performed within the union of all **M**s [9] of *Gymnocalycium* species. Then, we parsed the full niche ranges into equal-sized discrete bins (S3 Table) separately for temperature (~1°C width) and for precipitation (~100 mm width). The BR (binned-range) character coding strategy embedded in the BAR method is as follows [9]: (1) All occurrences, and the bins that lie between minimum and maximum values of occurrence points within the area **M** are marked as "present" (code - 1), (2) If there are unoccupied conditions within the **M** beyond the limits of occupied conditions, as estimated in the previous step,

such bins are marked as "absent" (code - 0), and (3) If the environmental limits at either end of the occurrence information overlap with the boundary conditions of **M,** the environmental bins beyond these limits are marked as "uncertain" (code -?). In the binning scheme of species for which the **M** simulation failed due to a low number of records, the bins located beyond the lower and upper limits of the occupied environmental ranges were coded as 'uncertain' to incorporate the uncertainty associated with the climatic distribution of such species. Maximum likelihood reconstruction of ancestral niches was performed using the *nichevol* R package (version 0.1.20; https://github.com/marlonecobos/nichevol.git) [101]. Based on the BR character codes for a given bin of environmental values, the niche evolution was assessed as follows [9,14]: (1) presence in ancestor and absence in descendent is identified as retraction, (2) absence in the ancestor and presence in the descendent is identified as expansion, and (3) maintenance of identical states in ancestor and descendant, and no indication of changes due to the uncertainty of environmental utilization in ancestor and/or descendent, is termed as stasis (i.e., no change). If the current niche of the descendent species exceeds the upper limit of its ancestral niche, the niche change is termed as 'expansion high', and if the current niche range exceeds the lower limit of its ancestral niche, it is 'expansion low'. Similarly, if the niche of the descendent species retracts from the upper limit of its ancestral niche, it is 'retraction high', and if the retraction occurs at the lower limit, it is 'retraction low'. For the currently-known species that experienced niche changes, ancestral niches, niche evolution, and current niches were represented in geographic space within the corresponding accessible areas. Boxplots were developed using the *ggplot2* R package [102] to represent the ranges of environmental values at which the evolution occurred.

## Results

### Nearly half of the taxa experience thermal niche evolution

We analyzed the utilization of the temperature variable by each of the species (i.e., the realized temperature niche) (S3 Table; S1 Fig.). At the lower-limit spectrum, *G. hossei* utilized the lowest temperature (1.5°C), and *G. stenopleurum,* the highest (24.3°C), whereas, on the higher-limit spectrum, the lowest temperature (14.2°C) was used by *G. reductum*, and the highest limit (25.4°C) was occupied by both *G. eurypleurum* and *G. stenopleurum*. *Gymnocalycium mesopotamicum* used the lowest range of temperature (0.5°C), and *G. marsoneri*, the widest (18.2°C). In the simulated accessible areas, the minimum width of temperature range was observed for both *G. mesopotamicum* and *G. stenopleurum* (1.1°C), and the maximum width was noted for *G. hossei* (29.5°C).

Nearly half of the species (17/40) in the *Gymnocalycium* clade experienced changes in temperature niches over time. Both expansion and retraction events were noted across the phylogeny (Fig 3). The thermal niches of *G. saglionis*, *G. ritterianum*, *G. bruchii*, *G. paraguayense*, and *G. mihanovichii* were only expanded with respect to that of their ancestors. The niche expansion events in *G. spegazzinii* and *G. schickendantzii* were also accompanied by niche retractions. No ancestral temperature niches were retained by *G. schickendantzii*. *Gymnocalycium castellanosii*, *G. rhodantherum*, *G. pugionacanthum*, *G. bodenbenderianum*, *G. mesopotamicum*, *G. calochlorum*, *G. striglianum*, *G. reductum*, *G. marsoneri*, and *G. pflanzii* experienced only niche retractions. With reference to the **M**s, we estimated the utilization of temperature niches by ancestors and descendants (S2 Fig.). The thermal niche expansion at the upper end of the ancestral niche limit was noted only for *G. paraguayense*, whereas, all other expansion events occurred at the lower limits. The upper limits of the ancestral niches were retracted for *G. rhodantherum*, *G. spegazzinii*, *G. reductum*, *G. striglianum*, and *G. pflanzii*; for the remaining species, niche retractions were at the lower limits, except for *G. mesopotamicum* and *G. marsoneri*, for which complete niche retractions with respect to the ancestral conditions were noted; however, for these species, the ancestral thermal niches corresponding to the current conditions were uncertain (Fig 3). The ancestral niches, niches of descendant species, and niche changes were represented in the geographic space (S3 Fig.). We noted that most of the upper limit retraction events were reflected geographically at the eastern side of the species' current distributions. Mixed geographic patterns were observed in other niche evolutionary events (i.e., expansions and lower limit retractions).

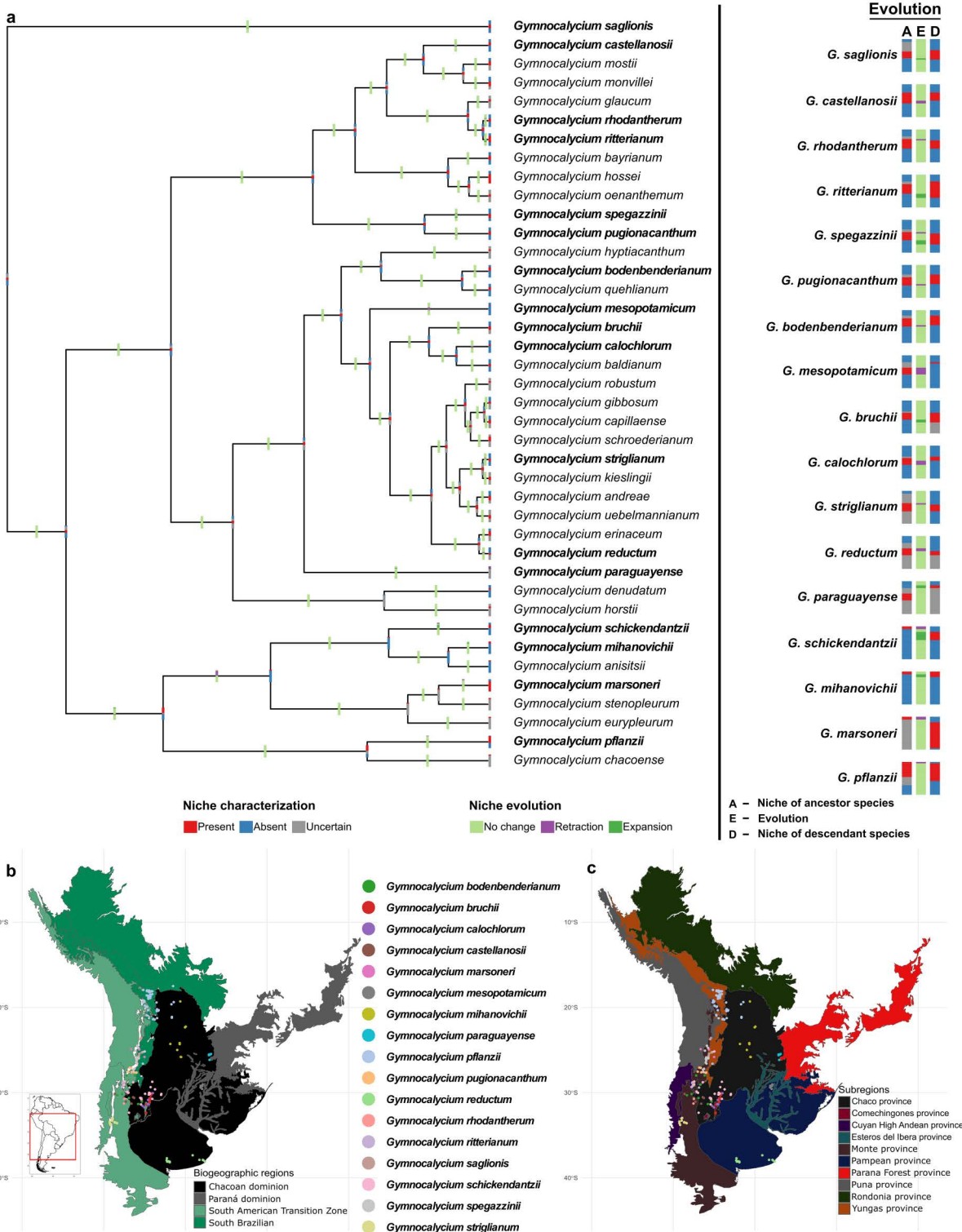

**Fig 3. Reconstructed temperature niches.** The upper panel shows the evolutionary dynamics of temperature niches across the *Gymnocalycium* phylogeny. The bottom panel indicates the bioregion-specific distribution of 17 *Gymnocalycium* species that experienced temperature niche evolution. The shapefiles of countries were sourced from DIVA-GIS (https://diva-gis.org), a free and open-source geographic information system.

## Precipitation niches are retracting

The utilization of the precipitation variable by each of the *Gymnocalycium* species is given in figure S4 (i.e., the realized precipitation niche) (S3 Table). The lowest value of the precipitation within the lower-limit spectrum was used by *G. bodenbenderianum* (108 mm), and the highest value was noted for *G. horstii* (1534 mm). *Gymnocalycium glaucum* tolerated the lowest value of the precipitation (206 mm), and the utilization of the highest precipitation value (1813 mm) was observed for *G. paraguayense*, within the higher-limit spectrum of the precipitation values. *Gymnocalycium glaucum* used the narrowest precipitation range (77 mm), and *G. schroederianum*, the highest (840 mm). In the simulated **M**s, the lowest range of precipitation values available for utilization was noted for *G. glaucum* (77 mm), and the widest range of values available for utilization was observed for *G. marsoneri* (1562 mm).

Precipitation niche evolution was noted in only ~23% (9/40) of the species in the *Gymnocalycium* clade (Fig 4). None of the species experienced niche expansion events. Niche retraction events were noted in *G. castellanosii*, *G. spegazzinii*, *G. hyptiacanthum*, *G. mesopotamicum*, *G. bruchii*, *G. schroederianum*, *G. andreae*, *G. reductum*, and *G. eurypleurum*. The upper limit of the ancestral precipitation niche was retracted only in *G. castellanosii*, and the lower limit retraction was noted in *G. reductum*. All other species experienced complete precipitation niche retractions (S5 Fig.); however, the ancestral precipitation niches corresponding to the current conditions were uncertain in those species (Fig 4). The geographic representation of the niche evolutionary events revealed that the retraction at the upper-limit caused range contraction at the eastern side of the species' current distribution (S6 Fig.).

Among the species that experienced niche evolutionary changes, five species, viz., *G. bruchii*, *G. castellanosii*, *G. mesopotamicum*, *G. reductum*, and *G. spegazzinii*, were noted for evolving niches in both environmental dimensions (Table 1; Figs 3–4).

## Bioregion-specific evolutionary trends are prevailing

The biogeographic classification of this region comprises three dominions: Chacoan dominion (CD), Paraná dominion (PD), South Brazilian dominion (SBD), and the South American transition zone (SATZ) (Table 1). No evolutionary changes in precipitation niches were detected for species, distributed either partly or fully, within the South Brazilian dominion (Table 1; Fig 4). Among these, *G. pflanzii* is the only species well distributed in the SBD that experienced thermal niche evolution. Other species, such as *G. marsoneri*, *G. saglionis*, and *G. schickendantzii*, also underwent changes in thermal niches but showed sparse distribution within the SBD. In contrast, other well distributed species within this dominion, such as *G. baldianum*, *G. bayrianum*, and *G. oenanthemum*, have conserved both their thermal and precipitation niches. *Gymnocalycium denudatum*, *G. hyptiacanthum* and *G. schroederianum* are partially distributed in the Paraná dominion. The thermal niches of these species have been phylogenetically conserved; however, the precipitation niches have evolved in *G. hyptiacanthum* and *G. schroederianum*. Most of the *Gymnocalycium* species (80%; 32/40) have their distribution either fully or partly in the Chacoan dominion. Except for *G. spegazzinii*, the remaining eight species that experienced changes in the precipitation niches were distributed either partly (*G. andreae*, *G. bruchii*, *G. castellanosii*, *G. hyptiacanthum*, and *G. schroederianum*), or fully (*G. eurypleurum*, *G. mesopotamicum,* and *G. reductum*) in the CD. Among the 17 species, for which the thermal niches have evolved, *G. calochlorum*, *G. mesopotamicum*, *G. mihanovichii*, *G. paraguayense*, and *G. reductum* were distributed only in the CD, and the distribution of *G. striglianum*, *G. pugionacanthum*, *G. rhodantherum*, and *G. spegazzinii* were restricted only to the SATZ. Thermal niche expansion at the upper limit of the ancestral niche was noticed only in *G. paraguayense* distributed in the CD. No strict patterns of niche evolution were observed with regard to the altitudinal gradients (S7 Fig.); however, the width of altitudinal gradient (WAG) for all species that experienced changes in the precipitation niches was noted to be ≤ 1500 m, and 78% (7/9) of the species with precipitation niche changes were distributed at altitudes less than 2000 m.

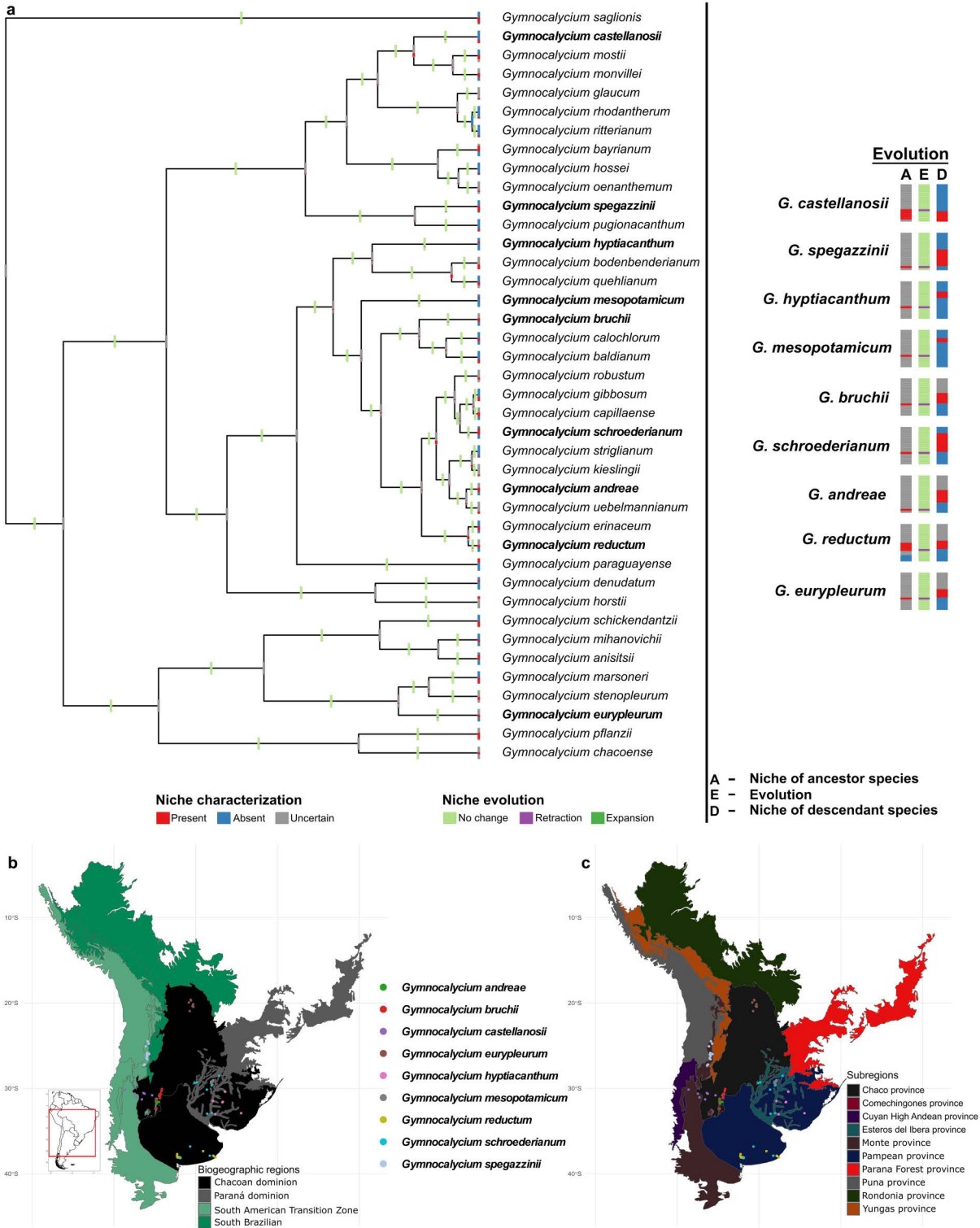

**Fig 4. Reconstructed precipitation niches.** The upper panel shows the evolutionary dynamics of precipitation niches across the *Gymnocalycium* phylogeny. The bottom panel indicates the bioregion-specific distribution of nine *Gymnocalycium* species that experienced precipitation niche evolution. The shapefiles of countries were sourced from DIVA-GIS (https://diva-gis.org), a free and open-source geographic information system.

Table 1. Bioregion-specific niche evolutionary trends in Gymnocalycium.

| S. No | Species | T | P | Biogeographic region |
|---|---|---|---|---|
| 1 | *Gymnocalycium andreae* | | ☑ | South American transition zone and Chacoan dominion |
| 2 | *Gymnocalycium anisitsii* | | | Chacoan dominion |
| 3 | *Gymnocalycium baldianum* | | | South American transition zone and South Brazilian dominion |
| 4 | *Gymnocalycium bayrianum* | | | South Brazilian dominion |
| 5 | *Gymnocalycium bodenbenderianum* | ☑ | | South American transition zone and Chacoan dominion |
| 6 | ***Gymnocalycium bruchii*** | ☑ | ☑ | South American transition zone and Chacoan dominion |
| 7 | *Gymnocalycium calochlorum* | ☑ | | Chacoan dominion |
| 8 | *Gymnocalycium capillaense* | | | South American transition zone and Chacoan dominion |
| 9 | ***Gymnocalycium castellanosii*** | ☑ | ☑ | South American transition zone and Chacoan dominion |
| 10 | *Gymnocalycium chacoense* | | | Chacoan and South Brazilian dominion |
| 11 | *Gymnocalycium denudatum* | | | Chacoan and Paraná dominions |
| 12 | *Gymnocalycium erinaceum* | | | South American transition zone and Chacoan dominion |
| 13 | *Gymnocalycium eurypleurum* | | ☑ | Chacoan dominion |
| 14 | *Gymnocalycium gibbosum* | | | South American transition zone, Chacoan dominion, and Andean region |
| 15 | *Gymnocalycium glaucum* | | | South American transition zone |
| 16 | *Gymnocalycium horstii* | | | Chacoan dominion |
| 17 | *Gymnocalycium hossei* | | | South American transition zone and Chacoan dominion |
| 18 | *Gymnocalycium hyptiacanthum* | | ☑ | Chacoan and Paraná dominions |
| 19 | *Gymnocalycium kieslingii* | | | South American transition zone and Chacoan dominion |
| 20 | *Gymnocalycium marsoneri* | ☑ | | South American transition zone, Chacoan, and South Brazilian dominion |
| 21 | ***Gymnocalycium mesopotamicum*** | ☑ | ☑ | Chacoan dominion |
| 22 | *Gymnocalycium mihanovichii* | ☑ | | Chacoan dominion |
| 23 | *Gymnocalycium monvillei* | | | South American transition zone and Chacoan dominion |
| 24 | *Gymnocalycium mostii* | | | South American transition zone and Chacoan dominion |
| 25 | *Gymnocalycium oenanthemum* | | | South Brazilian dominion |
| 26 | *Gymnocalycium paraguayense* | ☑ | | Chacoan dominion |
| 27 | *Gymnocalycium pflanzii* | ☑ | | South American transition zone, Chacoan, and South Brazilian dominion |
| 28 | *Gymnocalycium pugionacanthum* | ☑ | | South American transition zone |
| 29 | *Gymnocalycium quehlianum* | | | Chacoan dominion |
| 30 | ***Gymnocalycium reductum*** | ☑ | ☑ | Chacoan dominion |
| 31 | *Gymnocalycium rhodantherum* | ☑ | | South American transition zone |
| 32 | *Gymnocalycium ritterianum* | ☑ | | South American transition zone and Chacoan dominion |
| 33 | *Gymnocalycium robustum* | | | Chacoan dominion |
| 34 | *Gymnocalycium saglionis* | ☑ | | South American transition zone, Chacoan, and South Brazilian dominion |
| 35 | *Gymnocalycium schickendantzii* | ☑ | | South American transition zone, Chacoan, and South Brazilian dominion |
| 36 | *Gymnocalycium schroederianum* | | ☑ | Chacoan and Paraná dominions |
| 37 | ***Gymnocalycium spegazzinii*** | ☑ | ☑ | South American transition zone |
| 38 | *Gymnocalycium stenopleurum* | | | Chacoan dominion |
| 39 | *Gymnocalycium striglianum* | ☑ | | South American transition zone |
| 40 | *Gymnocalycium uebelmannianum* | | | South American transition zone and Chacoan dominion |

## Niche changes at the interface of paleoclimatic changes

Most of the diversification events in the *Gymnocalycium* clade occurred within the last one million years (0.03–0.95 Mya; S8 Fig.), coinciding with the Middle and Early Patagonian glaciations, however, two exceptions, *G. saglionis* (~3.79 Mya) and *G. paraguayense* (~1.45 Mya), diverged earlier, prior to the Great Patagonian Glaciations (GPG: 1.0–1.1 Mya).

These species experienced thermal niche changes only, with both showing evidence of niche expansion. All species that experienced evolution in their precipitation niches diverged in the last one million years. Within the spectrum of Early–Middle Pleistocene glaciations in this region, most of the niche changes in both temperature and precipitation dimensions, either retraction or expansion, occurred in the Post-GPG 2 phase (> 0.128 to < 0.710 Mya) (S8 Fig.). In the temperature dimension, the *Gymnocalycium* species that diversified during the Post-GPG 1 phase (> 0.710 to < 1.0 Mya) experienced either upper-limit retraction (*G. pflanzii*, ~ 0.95 Mya), or complete-non-utilization of environmental values (*G. schickendantzii*, ~ 0.78 Mya) with respect to that of their ancestors. Nonetheless, *G. schickendantzii* also experienced thermal niche expansion at the lower limit. Another species that diverged during Post-GPG 1 phase experienced complete-non-utilization of ancestral conditions, was *G. mesopotamicum*; however, the estimation of ancestral conditions for this species involved uncertainties. Half of the species (50%; 4/8) that diverged during the Post-GPG 2 phase underwent thermal niche retractions at the lower limits: *G. bodenbenderianum* (~0.20 Mya), *G. calochlorum* (~0.25 Mya), *G. pugionacanthum* (~0.50 Mya), and *G. castellanosii* (~0.51 Mya). A complete retraction of ancestral conditions with wide ranges of uncertainty in ancestral niches was noted in *G. marsoneri* (~0.39 Mya; Post-GPG 2). Either lower-limit expansion (*G. ritterianum*, ~ 0.04 Mya), or upper-limit retraction (*G. rhodantherum*, 0.04 Mya; *G. striglianum*, ~ 0.04 Mya; *G. reductum*, ~ 0.07 Mya) was observed in species that diverged during the Post-GPG 3 phase (Upper/Late Middle Pleistocene glaciations, < 0.128 Mya).

In the precipitation dimension, a complete-non-utilization of ancestral niches was noted in *G. hyptiacanthum* and *G. mesopotamicum* that diverged during the Post-GPG 1 phase. Similarly, a full retraction of ancestral niches was noted in the species diverged during the Post-GPG 2 phase, except for *G. castellanosii* that experienced upper limit retraction. A complete-non-utilization of ancestral conditions was noted in *G. andreae* (~0.09 Mya), and lower-limit retraction was observed in *G. reductum* (~0.07 Mya), diverged during the Post-GPG 3 phase. However, irrespective of the times of divergence, significant portions of the reconstructed ancestral precipitation niches of the species for which a complete-non-utilization of ancestral niches was noted were uncertain. Among the five species that experienced evolution in both temperature and precipitation dimensions, three diverged in the Post-GPG 2 phase: *G. bruchii* (~0.46 Mya), *G. spegazzinii* (~0.50 Mya), and *G. castellanosii* (~0.51 Mya).

## Discussion

### Retrospection: what, why and how?

Recent years have witnessed researchers using various methods and tools to reconstruct the ancestral niches [9]. The main shortfall of all such methods stems from the basic fact that it is difficult to estimate the fundamental niche of a species from the presence points alone [9]. Studies that used the summary statistics based on presence points alone pose higher risk of biased sampling, as the occurrence abundance of a species in a particular environment cannot always be interpreted that this suite of environmental conditions are highly favourable for the given species [9]. Because the observed abundance could be due to either biased sampling, or that it may be the most common environment in the accessible area. The BAR method stands out from the methods outlined [9] for two major reasons: the inclusion of the accessible area hypothesis in the analysis and the explicit incorporation of uncertainty. The methods that do not consider these two factors may produce results at the cost of over- or under-estimation of the true niche changes over time across the phylogeny [8]. The BAR method considers ranged response instead of a single value (median values of variables based on realized niche) which is used in more traditional generalized least squares reconstruction (GLS) approaches to estimate ecological niches. Hence, a potential limitation of the BAR method is the lack of an approach for the quantitative estimation of niche evolutionary rates that are comparable to the rates estimated from the conventional single-value reconstructions [9].

For the first time, our study reveals the evolutionary dynamics of the climatic niches of *Gymnocalycium* species in detail with respect to their ancestors, integrating a dated phylogeny. In plants, this approach (i.e., BAR method) was earlier used

by Ashraf et al. [16] to decipher the patterns of niche conservatism and evolution in the members of the Oleaceae family. Several other studies on plants, in which climatic niche evolution was investigated across phylogenies using different approaches [103–111], did not specifically incorporate an accessible area hypothesis for the extant species [7,9,83,84], and the inherent uncertainties associated with the incomplete characterization of fundamental niches (i.e., full niche ranges of extant species) [8,9]—the two interlinked and indispensable, yet often overlooked, components in the ancestral niche reconstruction workflows [9]. Hence, a clear depiction of the niche evolutionary dynamics (uncertainty, expansion, retraction, and/or stasis) in terms of the utilization of environmental values and the representation of such dynamics within the accessible areas of extant species, with respect to their ancestors, were also lacking in the above studies. In contrast, with the support of well-defined accessible area hypotheses for each of the *Gymnocalycium* species, we investigated four questions that paved the way for new valid insights on: 1) species' tendencies to conserve/evolve their niches over time, 2) variable-wise differences in the evolutionary dynamics, 3) bioregion specificities in the patterns of niche conservatism and/or evolution, 4) the role of paleo-climatic events on niche divergences, 5) niche evolution within the accessible areas, and 6) the need for species-specific and region-specific conservation efforts, by characterizing the niches of both extant species and their ancestors in two environmental dimensions (i.e., temperature and precipitation variables), specifically incorporating the 'uncertainty component'.

### Not very strict, but alarming

We observed clear evolutionary trends in both thermal and precipitation dimensions across the *Gymnocalycium* phylogeny; however, we also observed a relatively high, though not strict, conservation of precipitation niches. Cactaceae species are distributed mostly in arid environments, and their roots are specialized to extract water in precipitation-deficit conditions [112]. The roots of *Gymnocalycium* species experience evolution; basal lineages possess fascicular roots and terminal clades have napiform roots [69]. Fascicular roots facilitate the intake of water from shallow soils [113], and napiform roots help to store water to tolerate severe drought [114]. The tendency to conserve precipitation niche was more evident in species with fascicular roots than in those with napiform roots, as only three out of twenty-three species with fascicular root system experienced niche evolution with respect to precipitation dimension (viz., *G. castellanosii*, *G. eurypleurum,* and *G. spegazzinii*), whereas all remaining species that experienced changes in precipitation niche possess napiform root system [69]. We find this trend alarming because *Gymnocalycium* species with fascicular roots (see S1 Table for details) may be more sensitive to climate change, as superficial roots are highly sensitive to increases in soil temperatures [115,116]. In addition, climate change can cause these superficial roots to experience elevated respiration rates, and it may also result in reduced storage of water in shallow soil after rainfall events [112]. Hence, the populations of *Gymnocalycium* species with fascicular roots, that are distributed on the hot and dry limits of their ranges may face serious threats in a warming climate, leading to the contraction of their geographic ranges [112]. Non-adaptability to changing environments is the main reason behind the rise of species conservation issues, and such failures in adaptation largely represent cases of niche conservatism [117]. Ecophysiological studies that link the responses of these *Gymnocalycium* species to high temperatures may provide more insights on their thermal tolerances [118–120]. No expansion in precipitation niches further confirms the possibility of range contractions. Recently, Ringelberg et al. [121] reported conservation of precipitation niches across phylogeny in a large group of plants (Mimosoid legumes), and suggested that this phylogenetic niche conservatism is a major factor behind speciation in that group. Even so, we refrain from stating that the prevalence of high precipitation niche conservatism (>75%) is a major factor behind the speciation in *Gymnocalycium* group, as the evidence of niche conservatism across the phylogeny does not necessarily prove or disprove its role in lineage divergence [103] because speciation involves multiple stages [122].

### Bioregion-specific conservation practices

Bioregion-specific patterns were noted in the niche changes in *Gymnocalycium*. Based on the globally assigned aridity index (AI) values [123], the distributional patterns of *Gymnocalycium* species encompass arid, semi-arid, dry sub-humid,

and humid ranges (see [124], for details). The increase in humidity follows an eastward longitudinal pattern [124]. A high density of species composition exists in the western part of the current *Gymnocalycium* distribution, particularly in the SATZ, SBD, and CD, experiencing arid to semi-arid conditions. The *Gymnocalycium* species with conserved precipitation niches, possessing fascicular roots and distributed in these regions, may require special attention, as global warming can induce serious threats to such species assemblages. Not to overlook, the species distributed in the eastern part are not completely free from the threats, as global warming could cause a 56% expansion of dryland area by the end of this century [124,125].

## Pleistocene glaciations, speciation, and niche evolution

Freezing climatic conditions during the Pre-GPG, and three Post-GPGs in particular, might be the major driving force behind the lineage divergence events in the *Gymnocalycium* clade. In Cactaceae, a recent study [126] demonstrated that Pleistocene glaciations influenced speciation processes in the genus *Eriosyce*, using genetic data and species distribution models developed for present and past conditions. Cactus species are highly adapted to water-limiting condition [127]. Extreme cold conditions could affect the efficiency of cactus roots to extract water from soil, may be, in two ways: cell ruptures can occur due to the internal formation of ice crystals [128], and soil water conductivity becomes extremely low below 0°C [129]. These factors may be linked directly to why cactus distribution is restricted in freezing environments [130], although the lethality of extreme cold conditions does vary among different cactus species [131]. As an obvious non-preference for extreme cold conditions, the western limit of the current distribution of *Gymnocalycium* species is slightly shifted away from the eastern edges of the glaciers [132,133]. We assume that even though *Gymnocalycium* species are drought resistant, the surface water availability (shallow moisture content) could be a crucial driver in their distribution, underpinned by the observation that most species tend to conserve their precipitation niches. This is further supported by the facts that small precipitation pulses translate to shallow soil moisture content [134], and the sensitivity of the shallow root system of cactus species to low precipitation events is a function that facilitate the development of main roots [112,135,136]. The tropical niche conservatism hypothesis [137,138] supports the fact that plants in the Neotropical region tend to conserve their niches over time [139]. Following the diversification events in the Early–Middle Pleistocene glaciations, we presume that the *Gymnocalycium* species might have followed a relatively 'more tight-less tight' pattern in retaining the ancestral niche characteristics related to the precipitation and temperature variables, respectively; probably as an ecological strategy for conserving the critical variable at the expense of the other [41,140]. Species tend to adopt ecological strategies against evolutionary pressures within their habitats, and such strategies are shaped by the environment [141]. The prevalence of both expansion and retraction events in the temperature niches (i.e., more labile), along with only retraction events in the precipitation niches, further strengthens our assumptions. In addition, a recent study revealed that the seed mass of *Gymnocalycium* species is related to climatic variables, precipitation in particular [142]. Although *Gymnocalycium* species are distributed in arid environments, their study found that the precipitation variable is important in the early stages of growth in *Gymnocalycium* species, adding support to the observed phenomenon of conservatism in the precipitation variable.

## Conclusion

Our study provides new insights into the evolutionary dynamics of climatic niches within the genus *Gymnocalycium*, highlighting both the stability and adaptability of these species in response to environmental changes over time. By examining the phylogenetic evolution of niches with robustly defined accessible area hypotheses, we have elucidated patterns of niche conservatism and variability in temperature and precipitation across bioregions, highlighting the unique vulnerability of *Gymnocalycium* species with fascicular roots. These results show that while these cacti tend to conserve their precipitation niches, they are likely to face significant challenges in a warmer climate, particularly species from arid regions with shallow root systems. Furthermore, Pleistocene glaciations appear to have shaped the evolutionary trajectories of

*Gymnocalycium*, contributing to lineage divergence and niche specialization. Finally, our results highlight the importance of conservation strategies that address bioregion-specific challenges, especially as climate change threatens to alter the distribution and survival of these ecologically distinctive cacti. For the *Gymnocalycium* species distributed mostly in the mountain regions, *in situ* conservation programs may not serve the long-term goal. It is not possible for these species to simply move up and down the mountains to mitigate the climate change impacts, as land availability and dispersal capabilities are limited. Most of these species utilize specific climatic bands along elevational gradients. Therefore, future research that focuses on predicting the potential distribution of *Gymnocalycium* species in different periods (e.g., 2040, 2070, and 2100) under different scenarios of greenhouse gas emissions will help decide the viability of species-based *in situ* conservation strategies, utility of *ex situ* conservation efforts, and possibility of translocating plants from their current habitat to potential distribution areas on a timely basis, considering the climatic impacts.

## Supporting information

**S1 Fig. Available and occupied temperature ranges Temperature ranges based on the accessible areas represent available conditions, and those based on occurrence records represent ranges occupied by the species.**
(PDF)

**S2 Fig. Graphical representation of temperature niche changes.** The niche evolutionary events (i.e., retraction and/or expansion) at the upper and/or lower limits with respect to the ancestral niches are represented within the accessible areas (**M**) of 16 *Gymnocalycium* species. No ancestral ranges were plotted for *G. paraguayense* and *G. schickendantzii*, as their ancestors did not utilize the temperature ranges within their **M**-areas. No graph was plotted for *G. mesopotamicum*, as the predicted retraction fell outside the available temperature limits of its **M**-area.
(PNG)

**S3 Fig. Geographic representation of temperature niche changes.** The predicted niche evolutionary events (i.e., retraction and/or expansion) are represented in geographic space. For each species, the ancestral niche, evolutionary changes, and present niche are depicted geographically within the accessible areas (**M**).
(PNG)

**S4 Fig. Available and occupied precipitation ranges.** Precipitation ranges based on the accessible areas represent available conditions, and those based on occurrence records represent ranges occupied by the species.
(PDF)

**S5 Fig. Graphical representation of precipitation niche changes.** The niche evolutionary events (i.e., retraction and/or expansion) at the upper and/or lower limits with respect to the ancestral niches are represented within the accessible areas (**M**) of six *Gymnocalycium* species. No graph was plotted for *G. mesopotamicum*, as the predicted retraction fell outside the available precipitation limits of its **M**-area.
(PNG)

**S6 Fig. Geographic representation of precipitation niche changes.** The predicted niche evolutionary events (i.e., retraction and/or expansion) are represented in geographic space. For each species, the ancestral niche, evolutionary changes, and present niche are depicted geographically within the accessible areas (**M**).
(PNG)

**S7 Fig. Altitudinal gradients.** The distribution of the *Gymnocalycium* species across the altitudinal gradient. The colors indicate the evolutionary specificities with respect to the temperature and precipitation dimensions. TNE - Temperature niche evolution, PNE - Precipitation niche evolution, and Both - Niche evolution in both dimensions.
(PNG)

**S8 Fig. Dated phylogenetic tree.** The Bayesian evolutionary tree representing the relationship between the *Gymnocalycium* species, and their divergence time estimates. The green numerals indicate the divergence time estimates of *Gymnocalycium* species. The branch colors represent the evolutionary specificities with respect to the temperature and precipitation dimensions. Divergence time estimates are in millions of years.
(PNG)

**S1 Table. Distributional information.** Details of species used in the present study, and corresponding root systems.
(XLSX)

**S2 Table. Parameter settings.** Parameterization of the **M** simulations in the grinnell R package. Simulations were performed for the current scenario (1979–2013).
(DOCX)

**S3 Table. Binning.** Descriptive statistics of temperature and precipitation variables, and associated bins.
(XLSX)

## Acknowledgments

The first author would like to thank the National Scientific and Technical Research Council (CONICET) of Argentina for providing a scholarship.

## Author contributions

**Conceptualization:** Rahul Raveendran Nair, Alicia N. Sérsic, Diego E. Gurvich.

**Data curation:** Rahul Raveendran Nair, Alicia N. Sérsic, Pablo H. Demaio, Solana B. Perotti, Diego E. Gurvich.

**Formal analysis:** Rahul Raveendran Nair, Pablo H. Demaio, Solana B. Perotti.

**Funding acquisition:** Rahul Raveendran Nair, Diego E. Gurvich.

**Investigation:** Rahul Raveendran Nair, Alicia N. Sérsic, Pablo H. Demaio, Solana B. Perotti, Diego E. Gurvich.

**Methodology:** Rahul Raveendran Nair, Solana B. Perotti, Diego E. Gurvich.

**Project administration:** Rahul Raveendran Nair, Alicia N. Sérsic, Diego E. Gurvich.

**Resources:** Rahul Raveendran Nair, Alicia N. Sérsic, Pablo H. Demaio, Solana B. Perotti.

**Software:** Rahul Raveendran Nair.

**Supervision:** Rahul Raveendran Nair, Alicia N. Sérsic, Diego E. Gurvich.

**Validation:** Rahul Raveendran Nair, Alicia N. Sérsic, Pablo H. Demaio, Solana B. Perotti, Diego E. Gurvich.

**Visualization:** Rahul Raveendran Nair, Alicia N. Sérsic, Solana B. Perotti, Diego E. Gurvich.

**Writing – original draft:** Rahul Raveendran Nair, Diego E. Gurvich.

**Writing – review & editing:** Rahul Raveendran Nair, Alicia N. Sérsic, Pablo H. Demaio, Solana B. Perotti, Diego E. Gurvich.

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
