## [Decision Letter · Decision Letter 0]

15 Jan 2025

PONE-D-24-56930‘More tight-less tight’ Patterns in the Climatic Niche Evolution of Gymnocalycium (Cactaceae): Were Pleistocene Glaciations a Prelude?PLOS ONE

Dear Dr. Nair,

Thank you for submitting your manuscript to PLOS ONE. After careful consideration, we feel that it has merit but does not fully meet PLOS ONE’s publication criteria as it currently stands. Therefore, we invite you to submit a revised version of the manuscript that addresses the points raised during the review process. **Both reviewers found that it was a well written and generally sound study.  However, both reviewers also honed in on the fact that no procedure for selecting the variables used in models (from among 19 WORLDCLIM variables available) is described, leaving the impression that the two variables included in models were chosen somewhat arbitrarily.   I agree that a formal variable selection procedure, or at least a strong and explicit justification for the variables included and excluded, seems critical.  They also identify a number of more minor issues, including clarifications that would make the manuscript stronger. **

We look forward to receiving your revised manuscript.

Kind regards,

Patrick R Stephens, Ph.D.

Academic Editor

PLOS ONE

**Journal Requirements:**

The financial support received from the British Cactus and Succulent Society (BCSS, UK) for the project titled “Evolutionary history of four Gymnocalycium species of the Sierras Pampeanas at intra- and interspecific level” helped in part to complete this study. Details of the project can be found at:  https://bcss.org.uk/evolutionary-history-of-four-gymnocalycium-species-of-the-sierras-pampeanas-at-intra-and-interspecific-level/. 

3. We note that Figures 1, 4b - 4c and 8b - 8c in your submission contain map images which may be copyrighted. All PLOS content is published under the Creative Commons Attribution License (CC BY 4.0), which means that the manuscript, images, and Supporting Information files will be freely available online, and any third party is permitted to access, download, copy, distribute, and use these materials in any way, even commercially, with proper attribution. For these reasons, we cannot publish previously copyrighted maps or satellite images created using proprietary data, such as Google software (Google Maps, Street View, and Earth). For more information, see our copyright guidelines: http://journals.plos.org/plosone/s/licenses-and-copyright.

We require you to either present written permission from the copyright holder to publish these figures specifically under the CC BY 4.0 license, or remove the figures from your submission:

a. You may seek permission from the original copyright holder of Figures 1, 4b - 4c and 8b - 8c to publish the content specifically under the CC BY 4.0 license.  

Reviewers' comments:

Reviewer's Responses to Questions

**Comments to the Author**

1. Is the manuscript technically sound, and do the data support the conclusions?

Reviewer #1: Yes

Reviewer #2: Yes

2. Has the statistical analysis been performed appropriately and rigorously? 

Reviewer #1: Yes

Reviewer #2: Yes

3. Have the authors made all data underlying the findings in their manuscript fully available?

Reviewer #1: Yes

Reviewer #2: Yes

4. Is the manuscript presented in an intelligible fashion and written in standard English?

Reviewer #1: Yes

Reviewer #2: Yes

5. Review Comments to the Author

**Reviewer #1: ** The study aims to reconstruct ancestral niches for species of the genus Gymnocalycium, focusing on abiotic factors such as temperature and precipitation. The research is well-founded and employs a robust methodology that effectively addresses the requirements for hypothesis testing. Overall, this work makes a valuable contribution to understanding the evolutionary dynamics of cacti and is well-suited for publication.

I have the following comments and suggestions:

1. Line 75: I would not classify this genus as having a narrow distribution. This characterization conflicts with the statements on lines 346–347. Please revise for consistency.

2. Lines 90–109: Although this critique pertains to style, I found the hypotheses and their justifications difficult to follow (I had to reread a couple of times). I recommend summarizing the hypotheses into two or three key points for clarity and focus. Additionally, consider restructuring the paragraph to present the hypotheses first, followed by their justifications, as this would improve readability.

3. Lines 141–144: While I agree with the relevance of the variables selected, I am curious why other bioclimatic variables were not included, or why uncorrelated variables were not retained after a PCA of all 19 bioclimatic variables available at Bioclim. For instance, seasonality has been identified as a key driver of Neotropical regionalization, yet Bio15 (precipitation seasonality) was omitted. Please provide a adicional justification for this selection.

4. Line 194: While the relaxed clock model is a reasonable choice, it might be insightful to test whether a birth–death tree prior would better fit this diverse genus with many narrow endemics. The choice of tree prior significantly influences phylogenetic reconstructions and, consequently, ancestral character reconstructions (in this case: niche). I recommend evaluating model fit by comparing the Birth-Death with alternatives like the Yule model, using tools such as path sampling or stepping-stone sampling in BEAST2 (see: https://www.beast2.org/path-sampling/). This approach would enhance the robustness of age estimates and lend greater confidence to the hypotheses tested.

5. Figures: The manuscript includes 10 figures, which may overwhelm readers. I suggest transferring at least six figures to the supplementary material or consolidating them into more synthetic, comprehensive figures.

**Reviewer #2: ** This is my first review of the manuscript entitled, “‘More tight-less tight’ Patterns in the Climatic Niche Evolution of Gymnocalycium (Cactaceae): Were Pleistocene Glaciations a Prelude?” This is a well-written and engaging paper in which the authors investigate the factors contributing to the niche evolution of a narrowly distributed cactus genus. Despite its restricted distribution and endemism, the genus appears to be a suitable model for this type of study, as different species occupy a range of heterogeneous environmental conditions. Studies like this are still scarce in the cactus family, and I believe this paper represents a valuable empirical contribution to the field. However, I have a few concerns that I would like the authors to address.

Major Comments

Selection of Variables. My primary concern relates to the selection of variables used to model niche evolution. The argumentation is based solely on results obtained from two variables, and the authors do not mention any tests to justify their selection. This raises questions about whether these variables truly represent the most relevant factors for niche evolution.

Specific Comments:

1. Lines 123-125: Were the analyses also performed without the extrapolated samples? I am concerned about the potential bias these samples could introduce to the model. Please provide more details about how many samples fall into this category.

2. Line 132: Relying on only two records for G. chacoense seems insufficient for robust modeling. Please address this limitation.

3. Lines 185-187: Please, provide more details about the molecular data used. Where the sequences could be found?

BEAST and BEAUTi Usage

4. Lines 200-201: It appears that the analysis was conducted in BEAST, while BEAUTi was used primarily for setting priors. Please clarify this in the manuscript.

Validation of Selected Variables

5. Lines 208-211: The manuscript lacks any mention of tests to validate the chosen variables for estimating niche evolution. I recommend conducting a test to confirm the relevance of these variables for the species' current niches in comparison to other available variables.

Figures and Legends

6. Figure 2: Could you associate each projection of accessible areas with a geographic map? It is challenging to identify the occurrence regions for each species in the current format.

7. Figure 3 Legend: The terms “utilization” and “utilized” may not be appropriate. Species do not actively “use” temperature ranges; instead, they exhibit ecological requirements that permit them to occupy these ranges. Please revise the language.

8. Figure 3: Overlapping the occurrence and accessible area data would make the figure more informative. This would allow for better visualization of how variables in accessible areas correspond to those in occurrence areas.

9. Figure 6: When is the expansion considered high or low? I suggest including a discussion of this point in the text.

Minor Comments

1. Lines 78-81: The sentence is unclear. Please rephrase for better readability.

2. Line 117: Replace “global biodiversity information facility” with “Global Biodiversity Information Facility (GBIF)” and provide a link to the database.

3. Abbreviation Consistency: Please check the abbreviation of the genus name throughout the manuscript for consistency.

4. Lines 415-417: This sentence is better suited for the introduction. Consider relocating it.

5. Lines 419-420: Where are the results of the dated phylogeny? While I understand this is not the main focus of the paper, a brief discussion of diversification times would add value.

6. PLOS authors have the option to publish the peer review history of their article (what does this mean? ). If published, this will include your full peer review and any attached files.

**Do you want your identity to be public for this peer review?** For information about this choice, including consent withdrawal, please see our Privacy Policy .

Reviewer #1: No

Reviewer #2: No

---

## [Author Response · Author response to Decision Letter 1]

23 Jan 2025

From

Rahul Raveendran Nair

Postdoctoral Researcher

IMBIV, CONICET

Argentina

To

Dr. Patrick R. Stephens

Academic Editor

PLOS ONE

Sub: Submission of revised MS (PONE-D-24-56930; ‘More tight-less tight’ Patterns in the Climatic Niche Evolution of Gymnocalycium (Cactaceae): Were Pleistocene Glaciations a Prelude?

Dear Patrick:

First of all, we would like to thank you very much for allowing us to revise the manuscript based on the reviewers’ suggestions. It is a pleasure to learn that our manuscript could be a nice contribution to the current understanding about the evolutionary dynamics of cacti.

We really value the time and effort that the editor and the reviewers have taken to evaluate the manuscript. We have addressed all comments and suggestions meticulously. One of the major concerns was the choice of the variables. To clarify, we have provided a detailed justification for the selection of variables to the reviewers. We would like to mention the same here also for your kind perusal.

We understand the significance of choosing non-correlated (independent) variables via either PCA, or variable-elimination through correlation analyses, or through methods such as VIF etc., in conventional ecological niche modeling practices.

However, our focus here was different. Our attempt was to reconstruct the ancestral thermal and precipitation niches of Gymnocalycium species using the variables that most represent overall climatic distribution. Hence, we chose only Bio1 variable for the temperature dimension, and Bio12 variable for the precipitation dimension, to perform the analyses. This was not a random selection, and this decision was made during the conceptualization stage of the study. Because, these are the two important variables that reflect the overall climatic distribution of a species (Lin et al. 2017; Castro-Insua et al. 2018). The Bio1 and Bio12 are considered to be the “standard climatic variables” (Lin et al. 2017) for studying the overall climatic distribution of a species, and hence, researchers have directly used these variables to study the climatic niche evolution.

To avoid the possible confusion, we would like to mention here that we downloaded the full suite of climatic variables from PaleoClim database for employing them in simulating accessible areas (Area M) only, as an input for grinnell R package.

As the BAR method allows the characterization of niches in single dimension at a time, we preferred to study the niche conservatism/evolution using these two important variables (Bio1 and Bio12), that in general, reflect the critical aspects of the species’ overall climatic distribution (Castro-Insua et al. 2018; Wu et al. 2023).

1. Castro-Insua, A., Gómez-Rodríguez, C., Wiens, J.J. et al. Climatic niche divergence drives patterns of diversification and richness among mammal families. Sci Rep 8, 8781 (2018). https://doi.org/10.1038/s41598-018-27068-y

2. Lin, L.H, and Wiens J.J. Comparing macroecological patterns across continents: evolution of climatic niche breadth in varanid lizards. Ecography 40.8 (2017): 960-970.

3. Wu, G.L., Chen, D.X., Zhou, Z., Ye, Q., Baselga, A., Liu, H., Wen, Y. and Nong, S.Q., 2023. Climatic niche divergence explains angiosperm diversification across clades in China. Journal of Systematics and Evolution, 61(4), pp.698-708.

4. Owens, H.L., Ribeiro, V., Saupe, E.E., Cobos, M.E., Hosner, P.A., Cooper, J.C., Samy, A.M., Barve, V., Barve, N., Muñoz‐R, C.J. and Peterson, A.T., 2020. Acknowledging uncertainty in evolutionary reconstructions of ecological niches. Ecology and Evolution, 10(14), pp.6967-6977.

We have incorporated an additional sentence to justify the selection, as follows:

Lines 281-283: “We selected bio1 and bio12, as these variables reflect the most important aspects of the overall climatic distribution of a species [58,99]”

All other comments were properly addressed and the responses were written separately for the reviewers (appended below). We believe that the manuscript is now improved considerably.

Once again, we would like to express our gratitude to you for spending your valuable time and effort on improving the manuscript.

Thanks

-Rahul

Responses to the reviewers’ comments

Reviewer #1:

• The study aims to reconstruct ancestral niches for species of the genus Gymnocalycium, focusing on abiotic factors such as temperature and precipitation. The research is well-founded and employs a robust methodology that effectively addresses the requirements for hypothesis testing. Overall, this work makes a valuable contribution to understanding the evolutionary dynamics of cacti and is well-suited for publication.

o We greatly appreciate the time and effort that the reviewer has taken to review this manuscript. We are grateful to the reviewer for sharing positive comments about the quality of our MS. It’s a great pleasure to learn the reviewer’s assessment that our work contributes significantly to the present understanding of the evolutionary dynamics of cacti.

• Line 75: I would not classify this genus as having a narrow distribution. This characterization conflicts with the statements on lines 346–347. Please revise for consistency.

o Thanks for the suggestion. As the reviewer suggested, we have modified the sentence as follows:

Lines 76-80:

“This genus is of particular interest for studying environmental niche evolution across phylogeny due to its (1) high level of endemism [29,31,32], (2) adaptation to heterogenous environmental conditions [29,33–36], and (3) presence at different ecological regions across a range of altitudinal gradients [21,29–32,37,38]”

• 2. Lines 90–109: Although this critique pertains to style, I found the hypotheses and their justifications difficult to follow (I had to reread a couple of times). I recommend summarizing the hypotheses into two or three key points for clarity and focus. Additionally, consider restructuring the paragraph to present the hypotheses first, followed by their justifications, as this would improve readability.

o We thank the reviewer for this valuable observation. As suggested, we have modified the said section as follows:

Lines 81-161:

In this study, we tested the evolutionary dynamics of climatic niches across the Gymnocalycium phylogeny, asking the following questions.

1. Are the climatic-niches of narrow-ranged Gymnocalycium species highly conserved across the phylogeny, indicating niche conservatism?

Various well-established hypotheses, such as niche breadth-range size [39,40], range size-niche breadth [41–43], and range shift-niche breadth [44], that connect niche breadth with the size of geographic ranges, suggest a positive association between niche breadth and geographic range size [45–49]. In a phylogenetic framework, studies on climatic-niche evolution linked with range size suggest that the rate of niche evolution in narrow-ranged species may be lower (i.e., more conserved), compared to wide-ranged species [11,50–52]. However, inter-specific variations in the utilization of environmental space from the total environmental space of the clade may result in the divergence of niches [50]. There are inter-specific differences in the utilization of environmental conditions by the Gymnocalycium species. Hence, we expect that the climatic-niches of narrow-ranged Gymnocalycium species may not be highly conserved across the phylogeny.

2. Do the evolutionary dynamics of thermal and precipitation niches across the Gymnocalycium phylogeny follow similar patterns?

As there are strong selective pressures of temperature and precipitation variables on Gymnocalycium species distribution [29,30,33,37,38,53], we expect that the evolutionary dynamics of thermal and precipitation niches across the Gymnocalycium phylogeny do not follow similar patterns.

3. Does a bioregion-specific pattern of niche evolution exist with respect to temperature and precipitation dimensions?

Temperature and precipitation variables are central in biogeographic regionalization [54], and the Neotropics have diverse biogeographic regions (bioregions) [55,56]. Hence, we expect a bioregion-specific pattern of niche evolution in Gymnocalycium.

4. Did the Early-Middle Pleistocene glaciations (i.e. GPG and three Post-GPG phases) potentially drive the patterns of lineage divergence in Gymnocalycium species, resulting in the evolution of temperature and precipitation niches?

Several studies have linked the evolution of climatic niches with lineage divergence events [14,16,57–61]. In South America, the Great Patagonian Glaciation (GPG; 1.0-1.1 Mya) followed by three Middle Pleistocene glaciations (Post-GPG 1, 2 & 3) [62,63] greatly influenced species diversifications and distributions [64–67], as the glacial cycles of advancement and retreat had significant effects on the climate in southern South America (the regions south of 15oS) [68]. Hence, we expect that the Early-Middle Pleistocene glaciations (i.e. GPG and three Post-GPG phases) played a significant role in lineage divergence in Gymnocalycium species, leading to the evolution of temperature and precipitation niches.

• 3. Lines 141–144: While I agree with the relevance of the variables selected, I am curious why other bioclimatic variables were not included, or why uncorrelated variables were not retained after a PCA of all 19 bioclimatic variables available at Bioclim. For instance, seasonality has been identified as a key driver of Neotropical regionalization, yet Bio15 (precipitation seasonality) was omitted. Please provide additional justification for this selection.

o We clearly understand the reviewer’s concern.

We understand the significance of choosing non-correlated (independent) variables via either PCA, or variable-elimination through correlation analyses, or through methods such as VIF etc., in conventional ecological niche modeling practices.

However, our focus here was different. Our attempt was to reconstruct the ancestral thermal and precipitation niches of Gymnocalycium species using the variables that most represent overall climatic distribution. Hence, we chose only Bio1 variable for the temperature dimension, and Bio12 variable for the precipitation dimension, to perform the analyses. This was not a random selection, and this decision was made during the conceptualization stage of the study. Because, these are the two important variables that reflect the overall climatic distribution of a species (Lin et al. 2017; Castro-Insua et al. 2018). The Bio1 and Bio12 are considered to be the “standard climatic variables” (Lin et al. 2017) for studying the overall climatic distribution of a species, and hence, researchers have directly used these variables to study the climatic niche evolution.

To avoid the possible confusion, we would like to mention here that we downloaded the full suite of climatic variables from PaleoClim database for employing them in simulating accessible areas (Area M) only, as an input for grinnell R package.

As the BAR method allows the characterization of niches in single dimension at a time, we preferred to study the niche conservatism/evolution using these two important variables (Bio1 and Bio12), that in general, reflect the critical aspects of the species’ overall climatic distribution (Castro-Insua et al. 2018; Wu et al. 2023).

Castro-Insua, A., Gómez-Rodríguez, C., Wiens, J.J. et al. Climatic niche divergence drives patterns of diversification and richness among mammal families. Sci Rep 8, 8781 (2018). https://doi.org/10.1038/s41598-018-27068-y

Lin, L.H, and Wiens J.J. Comparing macroecological patterns across continents: evolution of climatic niche breadth in varanid lizards. Ecography 40.8 (2017): 960-970.

Wu, G.L., Chen, D.X., Zhou, Z., Ye, Q., Baselga, A., Liu, H., Wen, Y. and Nong, S.Q., 2023. Climatic niche divergence explains angiosperm diversification across clades in China. Journal of Systematics and Evolution, 61(4), pp.698-708.

Owens, H.L., Ribeiro, V., Saupe, E.E., Cobos, M.E., Hosner, P.A., Cooper, J.C., Samy, A.M., Barve, V., Barve, N., Muñoz‐R, C.J. and Peterson, A.T., 2020. Acknowledging uncertainty in evolutionary reconstructions of ecological niches. Ecology and Evolution, 10(14), pp.6967-6977.

As the reviewer suggested, we have incorporated an additional sentence to justify the selection, as follows:

Lines 281-283:

“We selected bio1 and bio12, as these variables reflect the most important aspects of the overall climatic distribution of a species [58,99]”

• 4. Line 194: While the relaxed clock model is a reasonable choice, it might be insightful to test whether a birth–death tree prior would better fit this diverse genus with many narrow endemics. The choice of tree prior significantly influences phylogenetic reconstructions and, consequently, ancestral character reconstructions (in this case: niche). I recommend evaluating model fit by comparing the Birth-Death with alternatives like the Yule model, using tools such as path sampling or stepping-stone sampling in BEAST2 (see: https://www.beast2.org/path-sampling/). This approach would enhance the robustness of age estimates and lend greater confidence to the hypotheses tested.

o Thanks for the suggestion. We really appreciate the reviewers’ effort to suggest other possibilities.

We chose Yule tree prior as our dataset involves DNA sequences of different species within the Gymnocalycium genus. This tree prior has been widely recommended to study speciation processes. If the sequence data is sufficiently informative, it has been recently reported that the choice of tree prior and molecular clock does not significantly affect the phylogenetic divergence estimates (Sarver et al. 2019).

Sarver BAJ, Pennell MW, Brown JW, Keeble S, Hardwick KM, Sullivan J, Harmon LJ. 2019. The choice of tree prior and molecular clock does not substantially affect phylogenetic inferences of diversification rates. PeerJ 7:e6334 https://doi.org/10.7717/peerj.6334

We believe that our sequence data is sufficiently informative, as the same molecular data was used earlier to study the species relationship within the Gymnocalycium genus (Demaio et al. 2011).

Demaio, P. H., Barfuss, M. H., Kiesling, R., Till, W., & Chiapella, J. O. (2011). Molecular phylogeny of Gymnocalycium (Cactaceae): assessment of alternative infrageneric systems, a new subgenus, and trends in the evolution of the genus. American Journal of Botany, 98(11), 1841-1854.

The first author of the above-publication (Pablo H. Demaio) is a co-author of this manuscript.

Hence, we prefer to move forward with the Yule tree prior in this manuscript.

5. Figures: The manuscript includes 10 figures, which may overwhelm readers. I suggest transferring at least six figures to the supplementary material or consolidating them into more synthetic, comprehensive figures.

o Thanks for this suggestion. We understand the reviewer’s concern. As suggested, six figures (Figures 3, 5, 6, 7, 9, and 10) were shifted to the supplementary section. Only four figures are placed in the main text.

Reviewer #2:

• This is my first review of the manuscript entitled, “‘More tight-less tight’ Patterns in the Climatic Niche Evolution of Gymnocalycium (Cactaceae): Were Pleistocene Glaciations a Prelude?” This is a well-written and engaging paper in which the authors investigate the factors contributing to the niche evolution of a narrowly distributed cactus genus. Despite its restricted distribution and endemism, the genus appears to be a suitable model for this type of study, as different species occupy a range of heterogeneous environmental conditions. Studies like this are still scarce in the cactus family, and I believe this paper represents a valuable empirical contribution to the field. However, I have a few concerns that I would like the authors to address.

o We greatly appreciate the time and effort that the reviewer has taken to review this manuscript. We are really thankful to the reviewer for sharing positive observations about the manuscript such that the manuscript could be a valuable empirical contribution to the current understanding of the evolutionary dynamics of cacti.

Major Comments

• Selection of Variables. My prima

---

## [Decision Letter · Decision Letter 1]

31 Mar 2025

PONE-D-24-56930R1‘More tight-less tight’ Patterns in the Climatic Niche Evolution of Gymnocalycium (Cactaceae): Were Pleistocene Glaciations a Prelude?PLOS ONE

Dear Dr. Nair,

Thank you for submitting your manuscript to PLOS ONE. After careful consideration, we feel that it has merit but does not fully meet PLOS ONE’s publication criteria as it currently stands. Therefore, we invite you to submit a revised version of the manuscript that addresses the points raised during the review process.

We look forward to receiving your revised manuscript.

Kind regards,

Patrick R Stephens, Ph.D.

Academic Editor

PLOS ONE

Journal Requirements:

**Additional Editor Comments:**

First off my apologies for how long this review process took.  One of the original reviewers declined to consider the revision, and it took considerable time to secure a new second reviewer.  The new reviewer has a number of suggestions related to clarity, and also suggests some new topics that may be worth addressing.  I agree with the reviewer that the Binned ancestral range reconstruction (BAR) method will likely still be unfamiliar to many readers, and so articulating the advantage of this approach, as well as the trade-off it perhaps represents, over the more familiar method of simply using maximum likelihood and ignoring uncertainty would help readers understand why you chose this method.  The discussion in Owens et al (2020) might be helpful for this. 

Owens, H. L., Ribeiro, V., Saupe, E. E., Cobos, M. E., Hosner, P. A., Cooper, J. C., ... & Peterson, A. T. (2020). Acknowledging uncertainty in evolutionary reconstructions of ecological niches. Ecology and Evolution, 10(14), 6967-6977.

 Reveiwer two also notes:

"The model appears oversimplified, as the used parameters assumes fixed numbers of events and dispersers across all species. Is this assumption in supported by empirical data for each species? Do all Gymnocalycium species have the same dispersal mechanisms and exhibit the same dispersal rates? Please explain these parameters choice and provide an evaluation of how potential variations in dispersal rates might influence the results."

It is not entirely clear to me whether the way that you have implemented BAR is as rigid as the reviewer implies.  My reading of the method is that it does not require such strict assumptions.  However, this comment does indicate either a methodological detail that needs to be better justified or a potential misunderstanding that you should head off in the minds or readers by being more clear  about how this unfamiliar method works.   I believe that the keywords you have chosen are appropriate, and I see no issue with them overlapping with the title.  However, feel free to retain them, change them or add additional key words as you wish. 

Overall this should be a fairly minor revision for the sake to clarity.  However, the issues raised by reviewer two seem substantial enough to me that I will likely send the revision back out for a third round of review.  That said, since one reviewer has already signed on this draft I will not require two reviews of your revision.

Reviewers' comments:

Reviewer's Responses to Questions

**Comments to the Author**

1. If the authors have adequately addressed your comments raised in a previous round of review and you feel that this manuscript is now acceptable for publication, you may indicate that here to bypass the “Comments to the Author” section, enter your conflict of interest statement in the “Confidential to Editor” section, and submit your "Accept" recommendation.

Reviewer #1: All comments have been addressed

Reviewer #3: (No Response)

2. Is the manuscript technically sound, and do the data support the conclusions?

Reviewer #1: Yes

Reviewer #3: (No Response)

3. Has the statistical analysis been performed appropriately and rigorously? 

Reviewer #1: Yes

Reviewer #3: Yes

4. Have the authors made all data underlying the findings in their manuscript fully available?

Reviewer #1: Yes

Reviewer #3: Yes

5. Is the manuscript presented in an intelligible fashion and written in standard English?

Reviewer #1: Yes

Reviewer #3: Yes

6. Review Comments to the Author

Reviewer #1: This is the second time I have reviewed this manuscript, and I appreciate the authors' efforts in addressing my previous comments. All the suggestions I provided in the first round of review have been either fully incorporated or adequately justified. The manuscript has improved significantly, and I find it suitable for publication in its current form. I therefore recommend its acceptance.

Reviewer #3: I would like to thank the editor for the invitation to review this manuscript. I found it highly interesting and appreciated the opportunity to explore some approaches that were previously unfamiliar to me. I received a revised version of the manuscript ‘More tight-less tight’ Patterns in the Climatic Niche Evolution of Gymnocalycium (Cactaceae): Were Pleistocene Glaciations a Prelude?”, which includes tracked changes and a response from the authors to previous reviewer comments. While I did not participate in the initial review round, I have examined both the revised manuscript and the authors' responses. Several of my own questions have already been satisfactorily answered, and overall, the study presents a well-structured investigation into the ecological and evolutionary dynamics of Gymnocalycium species.

The authors provide a strong theoretical foundation for their hypotheses and employ the Binned Ancestral Range (BAR) method to reconstruct past climatic niches, allowing for direct comparisons between precipitation and temperature variables. The manuscript applies adequate methodology and contributes valuable insights, but certain aspects require further clarification and refinement. Below, I outline specific points that I encourage the authors to address to improve its the scientific rigor and practical implications.

Keywords: The keywords repeat terms from the title. I recommend replacing redundant keywords with additional relevant terms to improve the manuscript’s discoverability in databases

Methodology: This was my main concern about the manuscript: the construction of the dispersal model. While the authors employ a novel and sophisticated approach to define the species accessible area hypothesis, the model appears oversimplified, as the used parameters assumes fixed numbers of events and dispersers across all species. Is this assumption in supported by empirical data for each species? Do all Gymnocalycium species have the same dispersal mechanisms and exhibit the same dispersal rates? Please explain these parameters choice and provide an evaluation of how potential variations in dispersal rates might influence the results.

The R package “grinnell” is cited after the parameter description. I suggest moving it to the beginning of the paragraph to improve readability.

The versions or GitHub commit references for both R packages ("grinnell" and "nichevol") are not provided. For reproducibility, please specify the exact versions used. Also, some methodological details are vague, such as the "default settings" used. mentioned without specifying which parameters were set by default. Please clarify the exact parameters to strengthen reproducibility. I recommend providing precise methodological details, possibly in a supplementary table, as done in Rojas-Soto et al. [2024 (doi 10.1111/jbi.14834)].

Results: The geographic and environmental distribution of Gymnocalycium (lines 350–355) is currently presented in the Results section, but this information primarily provides context rather than novel findings. I suggest moving this section to the Introduction, where it would serve as a clearer background for the study.

Discussion: The study finds that precipitation niches are largely conserved, while temperature niches show more flexibility. The interpretations are interesting, but the underlying drivers of this pattern remain somewhat speculative. As the study does not aim to directly test correlations between root systems and niche evolution patterns, it would be useful to discuss alternative explanations. Can the different rates of niche evolution be driven by other factors?

The authors suggests that Gymnocalycium species in certain bioregions can be at greater risk due to climate change, by connecting niche conservatism and extinction risk. Since this is a valuable perspective, I recommend discussing potential conservation strategies that could benefit from these findings. Additionally, since the study does not include ecological niche modeling (ENM) under future climate scenarios, I suggest briefly mentioning the importance of integrating ENM projections in future research to confirm species' vulnerability.

Conclusion: The phrase "In conclusion" is unnecessary and can be removed. The section already functions as a conclusion without explicitly stating it.

Biases and limitations: The study employs a Binned Ancestral Range (BAR) reconstruction to analyze niche evolution. While the authors present this as a robust approach, it remains relatively new and has been applied in only a limited number of studies. I encourage the authors to provide a more in-depth discussion of its potential limitations. How does BAR compare to other ancestral niche reconstruction methods in terms of potential biases? What are the main uncertainties associated with this method, and how might they affect the study's conclusions?

7. PLOS authors have the option to publish the peer review history of their article (what does this mean? ). If published, this will include your full peer review and any attached files.

**Do you want your identity to be public for this peer review?** For information about this choice, including consent withdrawal, please see our Privacy Policy .

Reviewer #1: No

Reviewer #3: No

---

## [Author Response · Author response to Decision Letter 2]

9 Apr 2025

Responses to the reviewer’s comments

Reviewer #3:

• I would like to thank the editor for the invitation to review this manuscript. I found it highly interesting and appreciated the opportunity to explore some approaches that were previously unfamiliar to me. I received a revised version of the manuscript ‘More tight-less tight’ Patterns in the Climatic Niche Evolution of Gymnocalycium (Cactaceae): Were Pleistocene Glaciations a Prelude?”, which includes tracked changes and a response from the authors to previous reviewer comments. While I did not participate in the initial review round, I have examined both the revised manuscript and the authors' responses. Several of my own questions have already been satisfactorily answered, and overall, the study presents a well-structured investigation into the ecological and evolutionary dynamics of Gymnocalycium species.

o We wholeheartedly thank the reviewer for having spent considerable time and effort in reviewing our manuscript by considering our previous responses to the reviewers. We appreciate the reviewer’s assessment that the manuscript details a well-structured investigation of the ecological and evolutionary dynamics of Gymnocalycium species.

• The authors provide a strong theoretical foundation for their hypotheses and employ the Binned Ancestral Range (BAR) method to reconstruct past climatic niches, allowing for direct comparisons between precipitation and temperature variables. The manuscript applies adequate methodology and contributes valuable insights, but certain aspects require further clarification and refinement. Below, I outline specific points that I encourage the authors to address to improve its scientific rigor and practical implications.

o Thanks for all these critical suggestions. We have addressed all your concerns. Necessary changes are incorporated in the manuscript at the appropriate places.

• Keywords: The keywords repeat terms from the title. I recommend replacing redundant keywords with additional relevant terms to improve the manuscript’s discoverability in databases

o Thanks for the suggestion. As the reviewer suggested, three redundant keywords were removed, and replaced with the words that are important in the context of our study. The keyword Gymnocalycium is retained because of its high significance related to the study. The current list of keywords includes Gymnocalycium, niche conservatism, precipitation, temperature, expansion, retraction.

• Methodology: This was my main concern about the manuscript: the construction of the dispersal model. While the authors employ a novel and sophisticated approach to define the species accessible area hypothesis, the model appears oversimplified, as the used parameters assumes fixed numbers of events and dispersers across all species. Is this assumption in supported by empirical data for each species? Do all Gymnocalycium species have the same dispersal mechanisms and exhibit the same dispersal rates? Please explain these parameters choice and provide an evaluation of how potential variations in dispersal rates might influence the results.

o We have understood the reviewer’s concern.

We used a simulation-based estimation of accessible areas for the 40 species of Gymnocalycium. The first version of this simulation was discussed in Barve et al. 2011, and it was based on fixed rules such that the movement is only allowed to neighboring cells. Later in 2021, Machado-Stredel et al. developed an extension to this method by incorporating a probability density function (dispersal kernel) that explains the probability of movement of dispersers to any position relative to the ‘starting population’. Then, the simulation of colonization and extinction events in cell grids occur based on the suitability of environmental conditions, assessed from a preliminary estimate of the fundamental ecological niche. These simulations were performed in several replications. In each replication, a subset of presence points would be kept as the ‘starting population’.

Barve, N., Barve, V., Jiménez-Valverde, A., Lira-Noriega, A., Maher, S.P., Peterson, A.T., Soberón, J. and Villalobos, F., 2011. The crucial role of the accessible area in ecological niche modeling and species distribution modeling. Ecological modelling, 222(11), pp.1810-1819.

Machado-Stredel, F., Cobos, M.E. and Peterson, A.T., 2021. A simulation-based method for selecting calibration areas for ecological niche models and species distribution models. Frontiers of Biogeography, 13(4).

The seed dispersal events in Gymnocalycium species are mainly mediated by the ants. As of now, no empirical studies on dispersal events have been reported in this genus so far. The two co-authors Pablo H. Demaio and Diego E. Gurvich have more than two decades of working knowledge in the field exploration of this genus. Their google scholar profiles are given below for the ease of verification, if necessary.

PHD: https://scholar.google.com.ar/citations?hl=en&user=fvX88eoAAAAJ

DEG: https://scholar.google.com/citations?user=RUNPpC8AAAAJ&hl=it

All simulated accessible areas were visually inspected by them to assess the plausibility of the accessibility of the current populations. Maximum dispersal events that can happen in 35 years are set to 35 (one per year) for all species. This fixed number (one per year) was chosen considering the two facts.

i) PHD and DEG have a strong conviction from their decades of field explorations that the dispersal events in Gymnocalycium can rarely exceed one in any of these species.

ii) As already mentioned above, no empirical studies on dispersal events have been conducted in Gymnocalycium so far.

However, despite all the above facts, we believe that empirical investigations on dispersal would be required to determine the accurate number of dispersal events in all these species.

We have specifically mentioned in the methodology that;

“Finally, the Ms that were simulated with an SD of 0.50 were chosen for further analyses (Fig 2). This selection was performed by superimposing all M polygons on the South American map, and choosing the appropriate Ms based on visual inspection and expert knowledge.” Getting realistic accessible areas that can be accessed by populations over a time span of 35 years was our prime interest.

From each colonized cell, dispersers’ movement per dispersal event will be based on the suitability of the cell. The suitability range (0-1) will be divided into ‘n’ number of intervals based on the maximum number of disperses; in this case, it is two. It indicates that per dispersal event, if the suitability of a cell (s) is < 0.5, only one disperser will move from that cell, and if the value of s > 0.5, two dispersers will move from that cell. Considering the present geographic distribution, and low levels of dispersal events per year, we chose to provide a platform where two levels of suitability (less suitable, s < 0.5 and more suitable, s > 0.5) were considered in each dispersal event from a colonized cell, and hence, the maximum number of dispersers was set to two.

For the ease of understanding, the following sentence is added under the sub-section ‘Accessible area hypothesis’ (Lines 187 – 192).

“The parameter (maximum number of dispersers) splits the suitability range (0-1) of a colonized cell into a number of intervals equal to the value of the parameter [84]. In this case, the suitability range (0-1) is divided into two equal intervals: 0.0–0.5 and 0.5–1.0. In each dispersal event, two dispersers will move from a colonized cell if the suitability of that cell exceeds 0.5, and only one disperser will move if the suitability is less than or equal to 0.5”.

Interested readers can get adequate information about these processes from Machado-Stredel et al. (2021).

• The R package “grinnell” is cited after the parameter description. I suggest moving it to the beginning of the paragraph to improve readability.

o As suggested by the reviewer, the sentence has been shifted up and placed along lines 181-183.

• The versions or GitHub commit references for both R packages ("grinnell" and "nichevol") are not provided. For reproducibility, please specify the exact versions used. Also, some methodological details are vague, such as the "default settings" used. mentioned without specifying which parameters were set by default. Please clarify the exact parameters to strengthen reproducibility. I recommend providing precise methodological details, possibly in a supplementary table, as done in Rojas-Soto et al. [2024 (doi 10.1111/jbi.14834)].

o As suggested following modifications were done to indicate the version and GitHub references for the packages “grinnell” and “nichevol”

“The M simulation procedure was performed using the grinnell R package (version 0.0.22; https://github.com/fmachados/grinnell.git) [84]”

“Maximum likelihood reconstruction of ancestral niches was performed using the nichevol R package (version 0.1.20; https://github.com/marlonecobos/nichevol.git) [101]”

Regarding the parameter settings, as suggested, a separate supplementary file is created as S2 Table, and the following sentence is added along lines 194–195.

“Parameterization of the M simulations is detailed in the supporting information (S2 Table)”

• Results: The geographic and environmental distribution of Gymnocalycium (lines 350–355) is currently presented in the Results section, but this information primarily provides context rather than novel findings. I suggest moving this section to the Introduction, where it would serve as a clearer background for the study.

o As suggested by the reviewer, the following sentence has been removed from the results and placed in the introduction along lines 76–79.

“The biogeographic classification of this region comprises three dominions: Chacoan dominion (CD), Paraná dominion (PD), South Brazilian dominion (SBD), and the South American transition zone (SATZ)”

• Discussion: The study finds that precipitation niches are largely conserved, while temperature niches show more flexibility. The interpretations are interesting, but the underlying drivers of this pattern remain somewhat speculative. As the study does not aim to directly test correlations between root systems and niche evolution patterns, it would be useful to discuss alternative explanations. Can the different rates of niche evolution be driven by other factors?

o Thanks for this suggestion. A recent publication (Bauk et al., 2025) has linked the precipitation variable with the seed mass of Gymnocalycium species, and suggested that the precipitation variable is significant in the early stages of growth in Gymnocalycium species. This is in support to our observation of the precipitation niche conservatism. Hence, we have added additional sentences in the ending of the last paragraph of the discussion along lines 572-577.

“In addition, a recent study revealed that the seed mass of Gymnocalycium species is related to climatic variables, precipitation in particular [142]. Although Gymnocalycium species are distributed in arid environments, their study found that the precipitation variable is important in the early stages of growth in Gymnocalycium species, adding support to the observed phenomenon of conservatism in the precipitation variable”

• The authors suggests that Gymnocalycium species in certain bioregions can be at greater risk due to climate change, by connecting niche conservatism and extinction risk. Since this is a valuable perspective, I recommend discussing potential conservation strategies that could benefit from these findings. Additionally, since the study does not include ecological niche modeling (ENM) under future climate scenarios, I suggest briefly mentioning the importance of integrating ENM projections in future research to confirm species' vulnerability.

o Thanks for this suggestion. As suggested, the following sentences have been incorporated at the end of the conclusion.

“For the Gymnocalycium species distributed mostly in the mountain regions, in situ conservation programs may not serve the long-term goal. It is not possible for these species to simply move up and down the mountains to mitigate the climate change impacts, as land availability and dispersal capabilities are limited. Most of these species utilize specific climatic bands along elevational gradients. Therefore, future research that focuses on predicting the potential distribution of Gymnocalycium species in different periods (e.g., 2040, 2070, and 2100) under different scenarios of greenhouse gas emissions will help decide the viability of species-based in situ conservation strategies, utility of ex situ conservation efforts, and possibility of translocating plants from their current habitat to potential distribution areas on a timely basis, considering the climatic impacts.”

• Conclusion: The phrase "In conclusion" is unnecessary and can be removed. The section already functions as a conclusion without explicitly stating it.

o Thanks for this suggestion. The phrase ‘In conclusion’ has been deleted.

• Biases and limitations: The study employs a Binned Ancestral Range (BAR) reconstruction to analyze niche evolution. While the authors present this as a robust approach, it remains relatively new and has been applied in only a limited number of studies. I encourage the authors to provide a more in-depth discussion of its potential limitations. How does BAR compare to other ancestral niche reconstruction methods in terms of potential biases? What are the main uncertainties associated with this method, and how might they affect the study's conclusions?

o Thanks for this suggestion. As suggested by the reviewer, the following paragraph is added in the beginning of the discussion.

“Recent years have witnessed researchers using various methods and tools to reconstruct the ancestral niches [9]. The main shortfall of all such methods stems from the basic fact that it is difficult to estimate the fundamental niche of a species from the presence points alone [9]. Studies that used the summary statistics based on presence points alone pose higher risk of biased sampling, as the occurrence abundance of a species in a particular environment cannot always be interpreted that this suite of environmental conditions are highly favourable for the given species [9]. Because the observed abundance could be due to either biased sampling, or that it may be the most common environment in the accessible area. The BAR method stands out from the methods outlined [9] for two major reasons: the inclusion of the accessible area hypothesis in the analysis and the explicit incorporation of uncertainty. The methods that do not consider these two factors may produce results at the cost of over- or under-estimation of the true niche changes over time across the phylogeny [8]. The BAR method considers ranged response instead of a single value (median values of variables based on realized niche) which is used in more traditional generalized least squares reconstruction (GLS) approaches to estimate ecological niches. Hence, a potential limitation of the BAR method is the lack of an approach for the quantitative estimation of niche evolutionary rates that are comparable to the rates estimated from the conventional single-value reconstructions [9].”

---

## [Editor Report · Decision Letter 2]

15 Apr 2025

‘More tight-less tight’ Patterns in the Climatic Niche Evolution of Gymnocalycium (Cactaceae): Were Pleistocene Glaciations a Prelude?

PONE-D-24-56930R2

Dear Dr. Nair,

We’re pleased to inform you that your manuscript has been judged scientifically suitable for publication and will be formally accepted for publication once it meets all outstanding technical requirements.

Kind regards,

Patrick R Stephens, Ph.D.

Academic Editor

PLOS ONE
---

## [Editor Report · Acceptance letter]

PONE-D-24-56930R2

PLOS ONE

Dear Dr. Nair,

I'm pleased to inform you that your manuscript has been deemed suitable for publication in PLOS ONE. Congratulations! Your manuscript is now being handed over to our production team.

Kind regards,

on behalf of

Dr. Patrick R Stephens

Academic Editor

PLOS ONE